# Effects of Spent Substrate of Oyster Mushroom (*Pleurotus ostreatus*) on Feed Utilization and Liver Serum Indices of Hu Sheep from the Perspective of Duodenal Microorganisms

**DOI:** 10.3390/ani14233416

**Published:** 2024-11-26

**Authors:** Mu-Long Lu, Guo-Hong Yuan, Chang-Chang Li, Li-Hong Hu, Xin-Wei Feng, Hui Jiang, Li-Lin Liu, Halidai Rehemujiang, Gui-Shan Xu

**Affiliations:** 1College of Animal Science and Technology, Tarim University, Alar 843300, China; a2774148614@163.com (M.-L.L.); 13309359388@163.com (G.-H.Y.); 18095365644@163.com (C.-C.L.); hx18699756542@163.com (L.-H.H.); fxwdky@126.com (X.-W.F.); jianghui308@126.com (H.J.); lll.net@163.com (L.-L.L.); 2Key Laboratory of Livestock and Forage Resources Utilization Around Tarim, Ministry of Agriculture and Rural Affairs, Tarim University, Alar 843300, China

**Keywords:** white-rot fungi, duodenal microorganisms, spent mushroom substrate, apparent digestibility, liver function

## Abstract

*Pleurotus ostreatus* spent mushroom substrate has high production but low utilization, and it is rarely fed to ruminants. This study evaluated the effects of the substrate on feed utilization and liver serum indices from the perspective of duodenal microorganisms. The results showed that the substrate had a complex effect on apparent digestibility, with the highest digestibility observed at the 10% replacement level. The addition of the substrate also influenced serum alanine aminotransferase levels. Although it did not affect the richness and diversity of duodenal microorganisms, the substrate was susceptible to contamination by *Trichoderma*, which could alter the intestinal microbiota structure. In conclusion, *Pleurotus ostreatus* spent mushroom substrate can be used in ruminant feeding, but its proportion should be controlled, with the optimal replacement level identified as 10% in this study.

## 1. Introduction

Spent mushroom substrate (SMS) is a byproduct of mushroom cultivation after several harvests and is primarily composed of mycelia, unutilized high-lignin substrates, and elevated levels of organic matter and enzymes [1]. Mushrooms are often referred to as ‘medicinal mushrooms’ due to their various beneficial properties, including immunomodulatory, antibacterial, cytostatic, and antioxidant effects, which have contributed to a growing demand over the years [2]. The output of SMS is nearly five times that of the mushrooms themselves, and with the rapid increase in mushroom consumption, SMS production has also surged [3]. SMS shows potential for various applications, such as serving as an organic fertilizer to enhance soil physicochemical properties and, in combination with pollutants, to mitigate environmental pollution [4]. When considering SMS as animal feed, safety remains a top priority. Although the nutritional content of SMS can vary based on different cultivation practices and the frequency of mushroom harvest, there is no significant impact on the levels of mycotoxins [1]. Despite the rising production of SMS and its numerous potential applications, only a small fraction is fully utilized, while the majority is either incinerated or improperly disposed, resulting in resource wastage and environmental pollution [5].

Previous studies have paid little attention to the use of SMS as animal feed. *Pleurotus* spp. exhibit a remarkable ability to selectively degrade lignin by producing various extracellular oxidizing enzymes [6]. This lignin degradation system enhances the accessibility of cellulose and hemicellulose for microbial digestion. Previous research has indicated that SMS is predominantly utilized as silage for ruminants, while direct feeding is less common. Yuan et al. [3] suggested that SMS could serve as an appropriate feed source for deer, as it demonstrated no negative effects on serum levels or feed conversion ratios. The growth performance and hematological parameters of growing sika deer showed no notable differences when the concentrate supplement in their diet was substituted with 10% to 20% of the P.SMS. However, a more substantial replacement, specifically substituting 30% of the concentrate supplements with the P.SMS, resulted in a significant reduction in hemoglobin (Hb) and hematocrit (HCT) levels [7]. In a study conducted by Agustinho et al. [8], SMS was ground and incorporated into whole-plant corn silage (WPCS) for fermentation. The results indicated that fermented WPCS did not affect the milk production performance of lactating goats; however, it did enhance the in vitro digestibility of the feed. In a separate study by Xu et al. [9], who evaluated the potential nutritive value of total mixed rations (TMR) with varying levels of SMS, it was found that 6.5% inclusion of SMS was the most beneficial. For monogastric animals, prior research has shown that small amounts of SMS can be used as a feed additive in the diets of weaned piglets, resulting in positive effects on both immunity and growth [10,11]. Our earlier study on Hu sheep indicated that the inclusion of P.SMS increased the feed-to-gain (F/G) ratio, which was explained from the perspective of rumen bacteria and fungi [12]. Collectively, these studies suggest that P.SMS, when included at appropriate levels, does not negatively affect animal health or performance across various species or feeding conditions. It may improve certain growth parameters and feed conversion rates while requiring careful attention to optimal inclusion levels.

The role of the rumen in ruminants is well-established; however, the small intestine of ruminants has also been shown to perform various essential functions [13,14]. Previous studies have underscored the critical role of the microbial consortium in the hindgut of ruminants in facilitating feed digestion and maintaining gastrointestinal tract health [15]. An in vitro study conducted by Li et al. [16] revealed that SMS could provide 36.27 g/kg of intestinal digestible crude protein, with the effective degradability of CP, ADF, and NDF being 36%, 23.54%, and 24.72%, respectively. Additionally, the gastrointestinal microbiota has been demonstrated to influence the host through various microbiome-gut-organ axes via its metabolites and interactions with microbial communities [15]. The portal vein in the liver acts as a conduit through which metabolites impact microorganisms and organs [17]. To the best of our knowledge, no study has focused on the effects of SMS as a feed ingredient on the intestinal flora of ruminants.

Given the potential benefits of SMS, it was used as a feed ingredient for Hu sheep in our studies. We hypothesized that P.SMS could be administered to sheep, positively contributing to their health and production. Our previous research focused on the rumen, where we discovered that a high percentage of P.SMS adversely affected production performance, rumen indices, and several serum indices in Hu sheep [12]. This negative impact was primarily attributable to the physical properties and substrate characteristics of P.SMS itself. Building on the potential benefits of SMS for lactating goats in digestibility [8] and the rumen changes in Hu sheep in our previous study, we speculated that P.SMS might also influence digestibility and the intestinal flora in the same group of Hu sheep. To explore this, we analyzed digestibility and serum liver indices to evaluate the impact of P.SMS. Additionally, we conducted sequencing of duodenal bacteria and fungi to investigate the effects of P.SMS on microbial communities. This approach provides a more comprehensive perspective on the potential utilization of P.SMS in Hu sheep, aiming to optimize its use.

## 2. Materials and Methods

### 2.1. Ethics Committee Approval

The sheep were purchased from Tumushuke Anxin Animal Husbandry Co., Ltd., Tumushuke, China. The animal study was reviewed and approved by the Animal Ethics Committee of Tarim University (approval numbers 2024049 and 2024069).

### 2.2. Spent Mushroom Substrate from Pleurotus ostreatus (P.SMS)

P.SMS was cultivated using a substrate consisting of 70% cottonseed hulls, 20% sawdust, 7% wheat bran, and 3% lime sourced from a mushroom plantation located on the outskirts of Alar City, Xinjiang. The chemical compositions of the P.SMS and WPCS used in the experiment are described in Table 1.

### 2.3. Experimental Animals and Group Design

A completely randomized experimental design was used in this study. Forty-five 3-month-old Hu male sheep with an average weight of 30.0 ± 1.2 kg were chosen and randomly assigned to 5 groups, with three replicates per group and three sheep per replicate. Each treatment group consisted of 9 sheep. These groups received varying levels of spent mushroom substrate (SMS) as treatments: Con (0%), PSMS5 (5%), PSMS10 (10%), PSMS15 (15%), and PSMS20 (20%). The basic diet, formulated to fulfill the requirements of a 30-kg sheep with a daily gain of 200 g, is detailed in Table 2 [18]. The diet was administered twice daily, once in the morning at 10:00 and again at 19:00. The adaptation period lasted for 10 days, followed by a 90-day trial period. Throughout the trial, all sheep were housed individually in a partially open sheep enclosure, provided with free access to salt blocks and clean water, and exposed to natural light and a cool environmental temperature. The average temperature and relative humidity were 18 °C and 28%, respectively.

### 2.4. Sample Collection and Processing

#### 2.4.1. Apparent Digestibility

On the 45th day of the positive test period, 30 sheep were randomly selected (2 sheep were randomly selected from each replicate). The selected sheep were housed in a single pen, where they were provided with free access to feed and water. The digestive test was conducted by total fecal collection, with a 6-day pre-test, followed by a 6-day positive test. Each sheep’s daily feed intake and amount of leftover feed were recorded during the positive test period. All feces from each sheep were collected and weighed daily. Ten percent of the daily fecal discharge from each sheep was mixed with 10% sulfuric acid for nitrogen fixation. The fecal samples were then stored at −20 °C for further evaluation.

Both the diet and fecal samples were dried at a constant temperature of 105 °C until their mass remained constant. The analysis of diet, ingredients, and fecal samples was conducted according to AOAC 930.15 for dry matter, AOAC 990.03 for crude protein (CP), AOAC 920.39 for ether extract (EE), AOAC 978.02 for Ca, and AOAC 946.06 for P. NDF and ADF contents were determined following Van Soest’s method [19]. The ME was calculated using the following formula: ME = 0.046 + 0.820 × (17.211 − 0.135 × NDF).

#### 2.4.2. Serum Biochemical Indices

Serum biochemical indices were collected from all 45 sheep on the 1st and 90th days of the experimental period before morning feeding using standard blood collection methods. Blood samples were collected in 5 mL tubes, which were allowed to stand for 40 min before centrifugation for 15 min. After centrifugation, the supernatant was transferred to 1.5 mL tubes for the determination of alanine aminotransferase, aspartate aminotransferase, alkaline phosphatase, total protein, and albumin concentrations using an automatic biochemical analyzer (Hitachi 7020 automatic analyzer, Hitachi Ltd., Tokyo, Japan).

#### 2.4.3. Intestinal Histomorphology

Immediately after the slaughter of all 45 sheep, the duodenum was isolated and rinsed with saline. A 3-cm segment of the intestine was excised from both the left and right sides of the middle section using surgical scissors. This segment was then trimmed to a length of 1 cm and placed in 4% paraformaldehyde solution for fixation.

The slices were prepared in the same manner as described by Huang et al. [20]. The prepared paraffin sections were examined under an electron microscope (Eclipse E200, Nikon, Shanghai, China), and images were captured using the microscope image processing system TL-507. Subsequently, the duodenal villus height, crypt depth, and thickness of the muscularis propria were measured using ImageJ software (version 1.4.10).

#### 2.4.4. High-Throughput Gene Sequencing

The duodenal samples of all 45 sheep underwent total genomic DNA extraction using the TGuide S96 Magnetic Soil DNA Kit (TIANGEN Biotech (Beijing) Co., Ltd., Beijing, China) following the manufacturer’s protocol. The quality and quantity of the extracted DNA were assessed by electrophoresis on a 1.8% agarose gel and quantified using a NanoDrop 2000 UV-Vis spectrophotometer (Thermo Scientific, Wilmington, NC, USA). For amplification of the full-length 16S rRNA gene, primer pairs 27F: AGRGTTTGATYNTGGCTCAG and 1492R: TASGGHTACCTTGTTASGACTT were utilized. The Internal Transcribed Spacer Identification (ITS) region was amplified using primer pairs ITS1-F: CTTGGTCATTTAGAGGAAGTAA and ITS4: TCCTCCGCTTATTGATATGC. Following quantification, amplicons with equimolar concentrations were pooled and sequenced on a PacBio Sequel II platform (Beijing Biomarker Technologies Co., Ltd., Beijing, China).

### 2.5. Operational Taxonomic Unit (OTU) Generation Process

The raw reads obtained from sequencing were subjected to filtering and demultiplexing using SMRT Link software (v8.0), with parameters set to minPasses ≥ 5 and minPredictedAccuracy ≥ 0.9 in order to obtain circular consensus sequencing (CCS) reads. Subsequently, Lima software (v1.7.0, github.com, accessed on 1 January 2023) was used to assign the CCS sequences to the corresponding samples based on their barcodes. CCS reads that were devoid of primers and those falling outside the length range of 1200–1650 bp were discarded, utilizing forward and reverse primer identification and quality filtering via Cutadapt (v2.7, [21]). The UCHIME algorithm (v8.1, drive5.com, accessed on 1 January 2023) was employed to detect and eliminate chimera sequences, resulting in the acquisition of clean reads. Sequences with a similarity greater than 97% were clustered into the same operational taxonomic unit (OTU) using USEARCH (v10.0, [22]), and OTUs with counts less than 2 in all samples were filtered out.

### 2.6. Statistical Analysis

The data concerning apparent digestibility, tissue morphology, blood parameters, microbial composition, and gene prediction, presented as average ± SEM, were collected and analyzed using one-way ANOVA in SPSS (v26.0). Blood parameters, also expressed as the mean ± SEM, were evaluated using two-way ANOVA in SPSS (v26.0), with treatment and time considered as two factors. Following the identification of significant differences, multiple comparisons were conducted using the LSD method, with a significance level set at *p* < 0.05.

Taxonomy annotation of the OTUs was conducted using the Naive Bayes classifier in QIIME2 [23] using the SILVA database [24] with a confidence threshold set at 70%. Alpha diversity was assessed to evaluate the complexity of species diversity within each sample using QIIME2 software (v2023.2). Beta diversity calculations were performed using non-metric multi-dimensional scaling (NMDS) to analyze species complexity across samples. Linear discriminant analysis (LDA) combined with effect size (LEfSe) was utilized to identify differentially abundant taxa [25], applying a threshold of *p* < 0.05 and LDA ≥ 2. To examine the influence on microbial communities, Random Forest analysis (v4.6-10) was executed using R (v3.1.1). Additionally, PICRUSt2 was employed to predict KEGG functions based on 16S rDNA data, while FUNGuild was used to predict functions based on ITS data.

## 3. Results

### 3.1. Effect of P.SMS on the Apparent Digestibility of Hu Sheep

Appendix A shows the average daily gain, dry matter intake, and feed-to-gain (F/G) ratio, which indicated that in our previous study [12], the PSMS20 group showed a significantly higher F/G ratio than the Con group (*p* < 0.05). Table 3 shows significant differences in the digestibility of Hu sheep among the Con, PSMS5, and PSMS10 groups in terms of Dry Matter (DM), Organic Matter (OM), and Crude Protein (CP). PSMS5 had lower digestibility of OM and CP than the Con and PSMS10 groups (*p* < 0.05). Furthermore, PSMS10 had a significantly higher digestibility of DM than PSMS5 (*p* < 0.05).

### 3.2. Effect of P.SMS on Duodenal Tissue Morphology

Table 4 shows the effects of feeding P.SMS on duodenal tissue morphology, while Figure 1 presents the paraffin-embedded duodenal sections. The results indicated no significant differences in tissue morphology, but an increasing trend in the V/C ratio was observed as the concentration of P.SMS increased (*p* = 0.080).

### 3.3. Effect of P.SMS on the Serum Liver Function Indices

Table 5 shows the serum liver function indices with time and treatment as factors, indicating that the inclusion of P.SMS affected serum Alanine aminotransferase (ALT) levels. The ALT levels in the PSMS5 group were significantly higher than those in the Con and PSMS20 groups (*p* < 0.05).

### 3.4. Sequencing Data Quality Assessment

The processing results of the sample sequencing data are shown in Appendix A. The sequencing results for bacterial and fungal samples revealed the following ranges for clean circular consensus sequences, the effectiveness of the sequence lengths, and the sequencing lengths: 6616–8510 bp for bacterial samples and 5282–8385 bp for fungal samples; effectiveness ranged from 97.02% to 99.58% for bacterial sequences and 97.02% to 99.30% for fungal sequences, with sequencing lengths of 1451–1463 bp and 582–654 bp, respectively.

### 3.5. Effect of P.SMS on Abundance, Composition, and Diversity of Duodenal Bacteria and Fungi

#### 3.5.1. OTU Composition and Species Distribution Composition

Sequences were clustered at a 97% similarity threshold, resulting in 1793 operational taxonomic units (OTUs) for bacteria and 470 OTUs for fungi across 25 samples (Figure 2A,B). Among the bacterial OTUs, 708 were common to all five groups, accounting for 39.49% of the total OTUs. The number of unique OTUs in Con, PSMS5, PSMS10, PSMS15, and PSMS20 was 15 (0.83%), 10 (0.56%), 45 (2.51%), 17 (0.95%), and 105 (5.86%), respectively. PSMS20 had the highest number of unique OTUs, while PSMS5 had the lowest. Among the fungi OTUs, the common OTUs of all five groups was 48, accounting for 10.21%. The number of unique OTUs in the five groups was 150 (31.91%), 38 (8.09%), 27 (5.74%), 47 (10.00%), and 24 (5.11%), respectively. The Con group had the highest number of unique OTUs, while PSMS20 had the lowest.

The dominant organisms at the phylum level of duodenal microbiota and fungi in all five groups of Hu sheep were Firmicutes and Ascomycota, with relative abundances ranging from 63.55% to 74.34% for Firmicutes and 64.86% to 86.11% for Ascomycota, respectively (Figure 2C,D). The Verrucomicrobiota and Spirochaetota bacterial phyla in PSMS20 were significantly higher compared to those in the Con group (Appendix A). Although changes in the abundance of the fungal phylum Neocallimastigomycota were evident in Figure 2D, there were no significant differences between fungal phyla across the five groups, according to Appendix A (*p* = 0.419). The highest abundance of Neocallimastigomycota was found in the PSMS5 group, with an abundance of 19.90%, and the lowest was in the PSMS20 group, with 0.01%.

In Figure 2E,F, the dominant organisms at the genus level in all five groups of Hu sheep were *Candidatus_Saccharimonas* in bacteria (with relative abundance ranging from 14.27% to 19.51%), and *Aspergillus* and *Trichoderma* in fungi (with relative abundances ranging from 22.34% to 38.95% and from 18.36% to 40.70%, respectively). As shown in Appendix A, the relative abundance of *Trichoderma* in the PSMS15 and PSMS20 groups tended to increase compared to that in the Con group (*p* = 0.057).

#### 3.5.2. Compositional Diversity Analysis of Bacteria and Fungi

Alpha diversity is shown in Figure 3, indicating no significant differences in diversity or uniformity among the five groups.

Figure 4 shows the stress values from both NMDS analyses below 0.1, indicating a reliable NMDS representation. The studies revealed no significant differences in the groups’ distribution of the bacterial and fungal microbiota. Additionally, Figure 4B shows that the points for different groups are closely clustered, suggesting similar microbial compositions across the groups.

### 3.6. Intergroup Variability and Associations in the Composition of Duodenal Bacterial and Fungi

LEfSe analysis revealed 43 differences among the five groups of duodenal bacteria and nine differences between the Con and PSMS15 groups of duodenal fungi. In Figure 5A,D, the phyla Bacteroidota and Rozellomycota were found to be significant between groups, with Bacteroidota ranking second and Rozellomycota third in Figure 6A,B, underscoring their importance as biomarkers. The relative abundance of Bacteroidota in the PSMS20 group was greater than that observed in other groups, as depicted in Figure 5B. Furthermore, Verrucomicrobia and Neocallimastigomycota were highly significant for sample classification, ranking first in Figure 6A,B, respectively. 

The genus *Lachnospiraceae_UCG_002* showed significant differences between groups in Figure 5A, with its relative abundance in the PSMS10 group being higher than that in the other groups, as shown in Figure 5C. Additionally, this genus ranked first in Figure 6C. The genus *Trichoderma* also demonstrated significance between the groups in Figure 5D, ranking first in Figure 6D, indicating its critical role as a biomarker. The relative abundance of *Trichoderma* in the PSMS15 group was higher than that in the other groups, as shown in Figure 5F.

### 3.7. Functional Gene Prediction on the Composition of the Duodenal Bacterial and Fungi

In Figure 7A and Appendix A, P.SMS had an impact on four out of the 20 pathways, including Amino Acid Metabolism, Cellular Community—Prokaryotes, Metabolism of Other Amino Acids, and Xenobiotics Biodegradation and Metabolism. Among these, PSMS15 exhibited a significantly higher abundance compared to Con, PSMS5, and PSMS10 groups. Although Figure 7B shows a noticeable change in the function of Animal Endosymbionts, no significant differences were found by one-way analysis of variance in Appendix A (*p* > 0.05).

## 4. Discussion

As the medicinal value of mushrooms has gained recognition, the production of mushroom substrates has also increased [2]. SMS is utilized in various ways but is often discarded or incinerated, leading to waste and environmental pollution [5]. Although the nutritional value of SMS varies depending on the composition of the mushroom substrate and the number of harvests, it remains a viable option as a feed ingredient [1]. Previous studies have demonstrated that the microbial consortium in the hindgut of ruminants plays a crucial role in both gastrointestinal tract health and feed digestion through metabolites [15]. Furthermore, it has been established that these metabolites can influence liver function via the portal vein [17]. In this experiment, P.SMS was employed as a substitute for WPCS, and its safety and feasibility as a feedstuff were assessed based on digestibility, liver indices, and changes in the intestinal flora.

The digestibility of various nutrients in the ration reflects an animal’s ability to break down and utilize these nutrients [26]. The dry material intake (DMI) and average daily gain (ADG) across the five groups exhibited no differences (Appendix A, *p* > 0.05), suggesting that the palatability of P.SMS was generally satisfactory and did not significantly influence sheep intake. However, when considering F/G indicators in Appendix A, the data for the Con group were lower than those for the PSMS5, PSMS10, and PSMS15 groups (*p* > 0.05) and significantly lower than those for the PSMS20 group (*p* < 0.05). This discrepancy is likely attributable to the difference in energy content between WPCS and P.SMS. In Table 2, ME was calculated using a prediction formula based on NDF, which suggests that P.SMS has a higher ME than WPCS. However, the actual results indicate that the energy content of P.SMS may be lower than that of WPCS. The increased DMI, coupled with a comparable ADG, may be driven by the lower energy content of P.SMS compared to WPCS, causing sheep to consume more dry matter to meet their energy requirements. This, in turn, leads to a significantly higher F/G ratio in the PSMS20 group compared to the Con group. Differences in F/G could also be the result of differences in apparent digestibility. Contrary to previous research by Long et al. [26], the addition of P.SMS appeared to decrease apparent digestibility in this study in certain proportions. Several factors can affect the apparent digestibility of feed in addition to the base nutrients of the feed, such as the NFC/NDF ratio [27]. Previous studies have indicated a negative relationship between particle size and digestibility. Specifically, grinding and reducing the particle size increases the feed’s specific gravity and passage rate, which subsequently reduces digestibility [28]. In this study, with the exception of PSMS10, the apparent digestibility of DM, OM, CP, NDF, and ADF in the other groups was lower than that observed in the Con group. This may be attributed to P.SMS providing a smaller particle size compared to WPCS. Beyond particle size, P.SMS may also enhance microbial opportunities through the lignin degradation system, potentially increasing the feed’s digestibility [6]. In this study, no differences were observed in the digestibility of ADF or NDF, which contradicts the assumption that substrates treated with white-rot fungi offer enhanced opportunities for microorganisms to degrade cellulose, hemicellulose, and other complex carbohydrates. Additionally, the nondigestible carbohydrates present in the mycelium of P.SMS may contribute to promoting intestinal health [4], potentially explaining the observed increase in the villus height to crypt depth ratio as the proportion of P.SMS increased (*p* = 0.080). Considering both the changes in digestibility associated with P.SMS and the improvements in intestinal morphology, a moderate ratio of P.SMS is likely to maximize feedstuff utilization. The digestibility of PSMS10 in this experiment may have resulted from a combination of these factors.

Bidirectional communication between the gut and the liver has been generally recognized in previous studies [29]. The ALT levels in the PSMS5 group were significantly higher than those in both the Con and PSMS20 groups (*p* < 0.05). This difference can be attributed to the treatments, as indicated by the two-way ANOVA (*p* < 0.05, Table 3), with no significant effect of time or interaction between treatments and time. In contrast, a study conducted by Yuan et al. [3], which administered SMS as feed to male sika deer at a replacement of 10%, demonstrated different serum biochemical indices to those observed in this study, suggesting that SMS did not affect liver function. The reasons for this discrepancy warrant further investigation.

Previous studies have indicated that a faster feed transit time, exposure to bile acids, and antimicrobial peptides influence intestinal microbial diversity and richness [30]. In this study, Firmicutes were the dominant bacteria, consistent with earlier findings in Aohan fine wool and Hu sheep [30,31,32,33]. However, in contrast to previous studies on Hu sheep, the abundance of Bacteroidota was not high; instead, Patescibacteria constituted the second largest share in this study [31,32,33]. The phylum Bacteroidota ranked second after Verrucomicrobiota in the random forest analysis, with its abundance ranging from 0.62% to 3.04%. Notably, PSMS5 exhibited the lowest abundance of Bacteroidota, as detailed in Appendix A. Firmicutes are recognized for their effectiveness in energy harvesting and play a crucial role in body immune response regulation [31]. Bacteroides genomes contain various proteins specialized for breaking down different plant polysaccharides and host glycans [34]. Gharechahi et al. [35] also noted that Bacteroidetes-associated species exhibited a high abundance of genes encoding debranching and oligosaccharide-degrading enzymes, whereas Firmicutes-associated species demonstrated richness in cellulases and hemicellulases. The composition of these two phyla did not differ among the five groups in this experiment but contrasted with the results obtained by previous authors [31,32,33] on Hu sheep, likely due to the feed, which has lower NFC and provides relatively fewer polysaccharides to Bacteroidota. The apparent digestibility of NDF and ADF did not exhibit significant differences, which aligns with the lack of variation among the Firmicutes. The newly identified bacterial phylum Verrucomicrobia is found in the intestinal mucosa of healthy individuals [31]. Additionally, it has been observed in the gut microbiota of several species, such as Tan sheep [36], blue sheep (Pseudois nayaur) [37], and Hu sheep [31,32]. Research indicates that Verrucomicrobia has anti-inflammatory effects and contributes to the regulation of glucose homeostasis in the host microbiome [38]. Patescibacteria, also known as candidate phyla radiation (CPR), constitute a diverse and notably substantial portion of microbial dark matter, much of which remains largely unidentified, unclassified, and underexplored [39]. Patescibacteria, previously referenced in studies involving Hu sheep intestines, was highlighted by Zhang et al. [33] for its ability to reduce NO_2_, NO, and N_2_O, thereby mitigating energy loss due to environmental changes. The genus *Candidatus_Saccharimonas* was noted by Ma et al. [30] as a unique genus of bacteria in the foregut of Aohan Fine-Wool Sheep. In this study, no significant difference in the abundance of *Candidatus_Saccharimonas* was observed between the five groups. *Lachnospiraceae* is an anaerobic bacterium that plays a role in dietary fiber degradation and produces short-chain fatty acids [40]. In a study conducted by Ma et al. [30], it was noted that *Lachnospiraceae* are the main components of the intestinal microbiota of ruminants. The family *Lachnospiraceae* was not identified as significant in Figure 5A, whereas several genera levels were identified as significant. Functional gene predictions for bacteria indicated a growing trend in amino acid metabolism with the addition of P.SMS. This observation further elucidates why PSMS10 exhibited the highest CP apparent digestibility.

Although less abundant than bacteria, fungi efficiently degrade fiber and play a crucial role in the initial colonization and disruption of feed particles [41]. It is commonly assumed that the primary function of fungi is to physically disrupt fibrous tissues during the early stages of feed particle colonization. In later stages, slow-growing fungi are believed to be outcompeted by fiber-adherent bacteria, which likely has a more significant impact on the pool of metabolites available to the host and, consequently, on efficiency metrics [41]. In this experiment, fungal diversity, abundance, and composition, as illustrated in Figure 2 and Figure 3, were similar regardless of the addition of P.SMS. This finding suggests that fungi may not play a significant role in the later stages of utilization, which is consistent with our hypothesis. In contrast to the rumen fungal composition, the relative abundance of Ascomycota was higher in the duodenum, with rumen relative abundance values ranging from 49.77% to 69.80% and duodenal relative abundance values ranging from 64.86% to 86.11% [12]. The increasing relative abundance of Ascomycota in the duodenum suggests its important role in this region. Conversely, the relative abundance of Neocallimastigomycota was reduced in the duodenal fungal composition of Hu sheep, with relative abundances ranging from 11.10% to 34.16% in the rumen and from 0.01% to 19.90% in the duodenum. Anaerobic fungi (phylum Neocallimastigomycota) are essential components of the gut microbiome in herbivores; however, they remain poorly characterized, and their biosynthetic enzymes, including those involved in the production of natural products such as antibiotics, have not been extensively studied [42]. In this experiment, the phylum Neocallimastigomycota exhibited variations between groups, but without significant differences (*p* > 0.05), and was ranked first in the random forest analysis. This study utilized generic fungal primers rather than specialized primers for Neocallimastigomycota, indicating that there may be subtle differences in other levels of biomarkers that warrant further investigation. At the genus level, *Aspergillus* and *Trichoderma* were identified as the dominant genera in the duodenum, consistent with our previous study on rumen fungal composition [12]. *Aspergillus* has demonstrated the potential for producing single-cell proteins and utilizing its organic residues [43]. In this study, no significant difference was observed in the relative abundance of *Aspergillus*. However, its high relative abundance percentage, ranging from 22.34% to 38.95% across all groups, warrants further investigation. Santiago et al. [44] noted that *Trichoderma* frequently contaminates mushroom substrates, leading to a decrease in mushroom production. We suggest that this phenomenon may be attributed to the P.SMS substrate, which, when affected by *Trichoderma*, resulted in an increase in the relative abundance of *Trichoderma* in the duodenum as the proportion of P.SMS increased (*p* = 0.057). This situation was also observed in our previous studies of rumen fungi in Hu sheep, although no significant difference was noted [12].

## 5. Conclusions

In this experiment, replacing 10% of WPCS with P.SMS resulted in improved digestibility. Contrary to our previous expectations, the addition of P.SMS did not significantly affect fiber digestion. Notably, the inclusion of 5% P.SMS had a significant impact on serum ALT levels, indicating the need for further investigation. Additionally, P.SMS notably influenced several intestinal flora and amino acid metabolism, highlighting the necessity for further research. The observed decrease in the abundance of Bacteroidetes may be attributed to the low NFC content in the diet, necessitating additional analysis to identify specific causes. Moreover, due to the susceptibility of P.SMS to contamination by *Trichoderma* during its production, an increase in the proportion of P.SMS was associated with a rising trend in the relative abundance of *Trichoderma* in the duodenum, which warrants further verification through additional studies.

## Figures and Tables

**Figure 1 animals-14-03416-f001:**
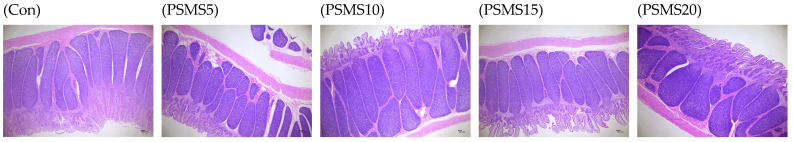
Duodenal tissue morphology of five groups. P.SMS replaced WPCS at levels of 0%, CON; 5%, PSMS5; 10%, PSMS10; 15%, PSMS15; or 20%, PSMS20.

**Figure 2 animals-14-03416-f002:**
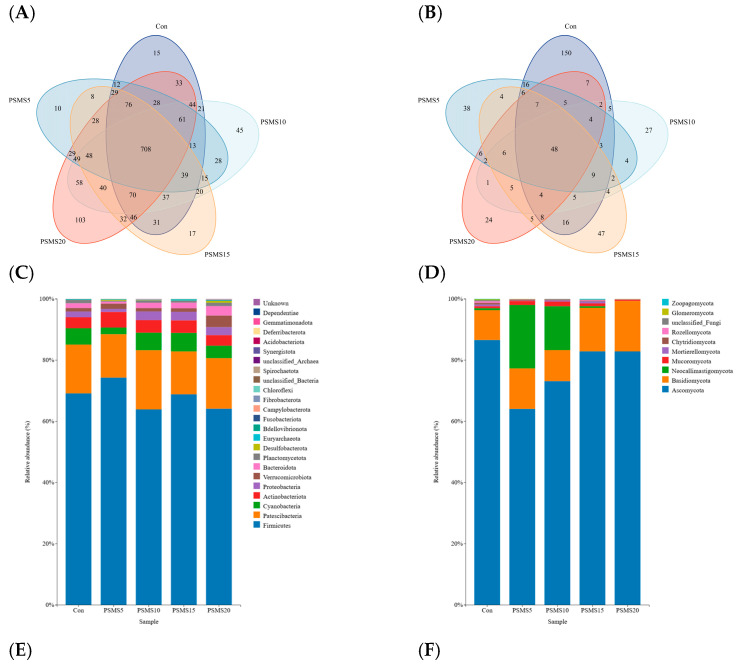
Venn diagram of OTUs in (**A**) bacteria and (**B**) fungi of Hu sheep. The relative abundance of duodenal (**C**) bacteria and (**D**) fungi at the phylum level. The relative abundance of duodenal (**E**) bacteria and (**F**) fungi at the genus level. P.SMS replaced WPCS at levels of 0%, CON; 5%, PSMS5; 10%, PSMS10; 15%, PSMS15; or 20%, PSMS20.

**Figure 3 animals-14-03416-f003:**
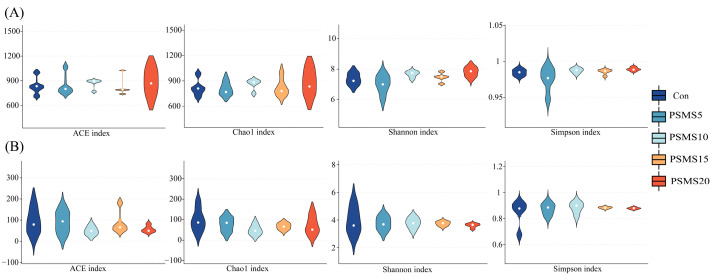
α diversity of bacteria (**A**) and fungi (**B**) samples. The four plots of the peers in (**A**,**B**) show the ACE, Chao1, Shannon, and Simson index for bacteria and fungi, respectively. P.SMS replaced WPCS at levels of 0%, CON; 5%, PSMS5; 10%, PSMS10; 15%, PSMS15; or 20%, PSMS20.

**Figure 4 animals-14-03416-f004:**
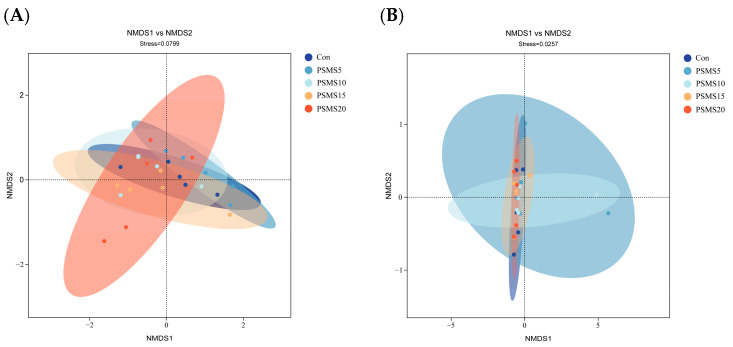
β diversity of (**A**) bacteria and (**B**) fungi samples. The dots in the figure indicate each sample, different colors represent different subgroups, and the distance between the dots indicates the degree of difference; when stress is less than 0.1, the data can be considered to have a good representation. P.SMS replaced WPCS at levels of 0%, CON; 5%, PSMS5; 10%, PSMS10; 15%, PSMS15; or 20%, PSMS20.

**Figure 5 animals-14-03416-f005:**
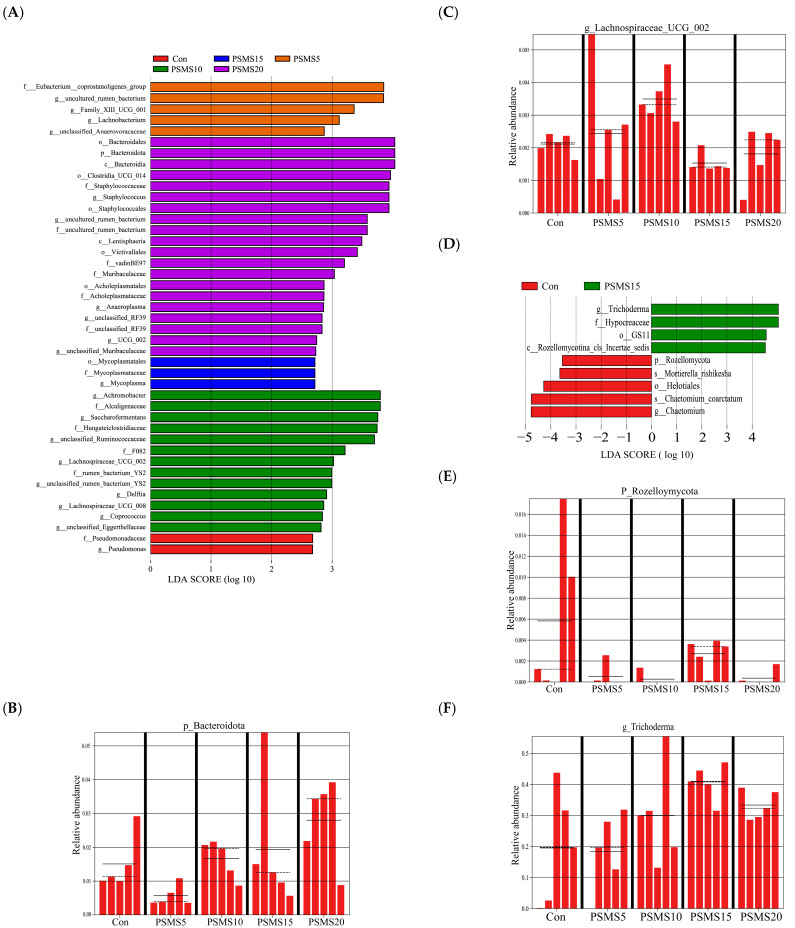
LEfSe analysis of bacterial and fungal samples. LDA analysis for (**A**) bacteria and (**D**) fungi shows the vertical coordinates as the categorical units with significant differences between groups, and the horizontal coordinates display the logarithmic score values of the LDA analysis for each categorical unit. Longer lengths on the horizontal axis indicate more significant differences. The threshold for LDA is set to 2. The distribution of the relative abundance of differential categories across different (**B**,**C**) bacterial and (**E**,**F**) fungal sample subgroups is shown, where solid and dashed lines represent the mean and median relative abundance of each categorical unit in each subgroup. P.SMS replaced WPCS at levels of 0%, CON; 5%, PSMS5; 10%, PSMS10; 15%, PSMS15; or 20%, PSMS20.

**Figure 6 animals-14-03416-f006:**
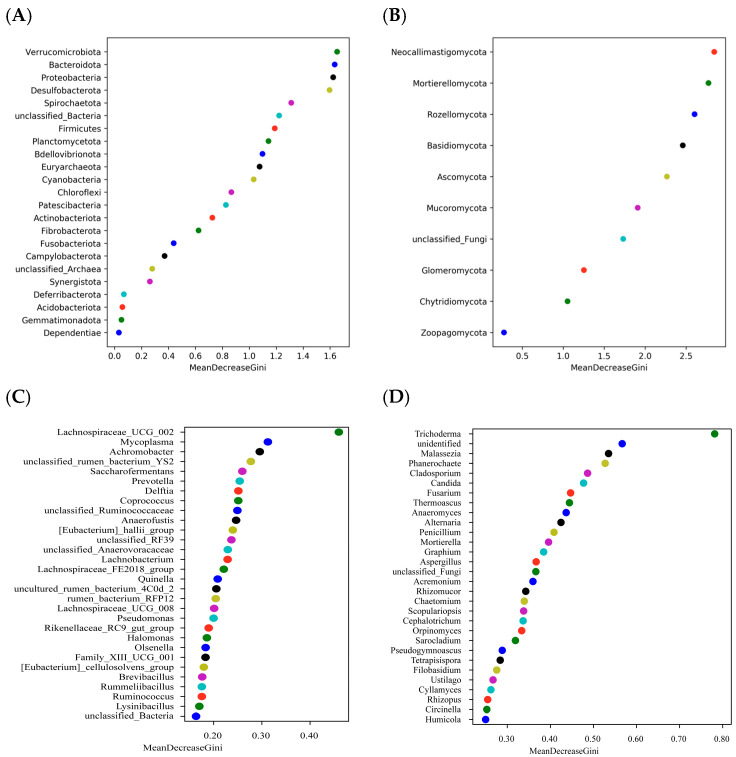
Random forest analysis of (**A**) bacteria and (**B**) fungi samples at the phylum level. Random forest analysis of (**C**) bacteria and (**D**) fungi samples at the genus level. The horizontal axis represents the species importance measure. Larger values on this axis imply higher importance of the species for sample classification, since the accuracy of sample classification decreases when the species is removed. The vertical axis shows the names of the species arranged in order of importance. Each point corresponds to both a specific species name on the left vertical axis and its associated species importance value on the horizontal axis. P.SMS replaced WPCS at levels of 0%, CON; 5%, PSMS5; 10%, PSMS10; 15%, PSMS15; or 20%, PSMS20.

**Figure 7 animals-14-03416-f007:**
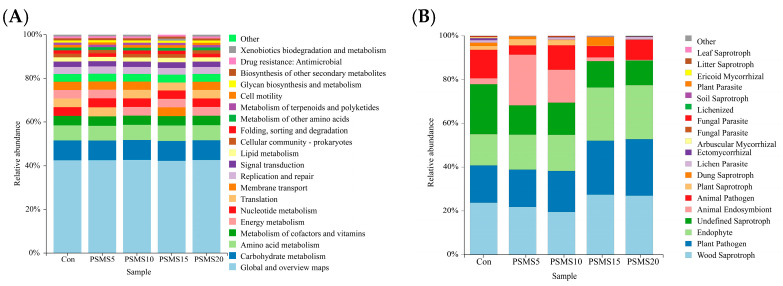
(**A**) Picrust2 Functionality Predictive Analytics for bacteria and (**B**) FunGuild Fungal Function Prediction. A bar graph displaying the functional composition of a group of bacteria or fungi using the histograms in the same column. P.SMS replaced WPCS at levels of 0%, CON; 5%, PSMS5; 10%, PSMS10; 15%, PSMS15; or 20%, PSMS20.

**Table 1 animals-14-03416-t001:** The chemical composition * of the P.SMS and WPCS used in the experiment.

Nutrient Levels ^†^	P.SMS Content (%)	WPCS Content (%)
Crude protein	8.74	6.77
Crude fat	2.41	4.40
Crude ash	29.41	13.39
Neutral detergent fibers	43.97	59.97
Acid detergent fibers	18.69	37.08
Calcium	0.63	0.59
Phosphorus	0.04	0.23

* The chemical composition was based on dry matter basis; ^†^ Nutrient levels were obtained from chemical analysis.

**Table 2 animals-14-03416-t002:** Test diet composition and nutrition level.

Items	Groups ^1^				
	Con	PSMS5	PSMS10	PSMS15	PSMS20
**Ingredients** **(%)**					
P.SMS	0.00	5.00	10.00	15.00	20.00
WPCS	52.36	47.36	42.36	37.36	32.36
Cornstalk	7.64	7.64	7.64	7.64	7.64
Corn	12.08	12.08	12.08	12.08	12.08
Wheat bran	6.32	6.32	6.32	6.32	6.32
Soybean meal	19.10	19.10	19.10	19.10	19.10
NaCl	1.00	1.00	1.00	1.00	1.00
Premix *	1.50	1.50	1.50	1.50	1.50
Total	100.00	100.00	100.00	100.00	100.00
**Nutrient levels** ^†^					
ME ^‡^ (MJ/kg)	9.10	9.18	9.27	9.36	9.44
CP	14.41	14.51	14.60	14.69	14.81
NDF	42.53	41.73	40.93	40.13	39.33
ADF	25.51	24.56	23.62	22.67	21.73
NFC	27.47	27.47	27.47	27.47	27.47
NFC/NDF	0.65	0.66	0.67	0.68	0.70
Ca	0.49	0.62	0.75	0.86	1.02
TP	0.38	0.38	0.37	0.37	0.36

^1^ P.SMS replaced WPCS at levels of 0%, CON; 5%, PSMS5; 10%, PSMS10; 15%, PSMS15; or 20%, PSMS20; * Premix provided the following per kg in basic diets: 10 mg of iron, 135 mg of manganese, 100 mg of zinc, 0.5 mg of cobalt, 12.5 mg of copper, 0.3 mg of selenium, 1.5 mg of iodine, 1400 IU of vitamin A, 500 IU of vitamin D, and 50 mg of vitamin E; ^†^ Nutrients were determined on a dry matter basis at 105 °C; ME, metabolic energy; CP, crude protein; NDF, neutral detergent fiber; ADF, acid detergent fiber; NFC, non-fibrous carbohydrate; TP, total phosphorus; ^‡^ The metabolic energy and non-fibrous carbohydrates in the nutritional level were calculated in reference to the Nutritional Requirements of Sheep for Meat in China [18], ME = 0.046 + 0.820 × (17.211 − 0.135 × NDF), and the rest were measured values.

**Table 3 animals-14-03416-t003:** Apparent digestibility of sheep in five groups.

Groups ^1^	Apparent Digestibility * (%)				
	DM	OM	CP	NDF	ADF
Con	65.03 ± 2.08 ^ab^	70.89 ± 3.54 ^a^	72.16 ± 3.95 ^a^	58.94 ± 3.69	44.25 ± 4.36
PSMS5	59.28 ± 3.24 ^b^	64.28 ± 3.65 ^b^	63.89 ± 2.54 ^b^	50.07 ± 4.56	42.89 ± 3.28
PSMS10	68.23 ± 2.48 ^a^	74.26 ± 3.89 ^a^	75.11 ± 3.69 ^a^	56.38 ± 3.98	43.56 ± 2.69
PSMS15	62.41 ± 4.36 ^ab^	66.43 ± 2.59 ^ab^	66.84 ± 2.14 ^ab^	54.52 ± 2.14	42.63 ± 3.59
PSMNS20	63.56 ± 2.56 ^ab^	67.59 ± 3.65 ^ab^	70.54 ± 2.39 ^ab^	51.68 ± 5.36	39.56 ± 4.36
*p*-value	0.048	0.039	0.046	0.896	0.122

^1^ P.SMS replaced WPCS at levels of 0%, CON; 5%, PSMS5; 10%, PSMS10; 15%, PSMS15; or 20%, PSMS20; * The labels ‘a’ and ‘b’ in the same row indicate significant differences between different groups (*p* ˂ 0.05). The absence of labels indicates no significant difference.

**Table 4 animals-14-03416-t004:** Effect of P.SMS on duodenal tissue morphology in Hu sheep.

Groups ^1^	Items *			
	Villus Height	Crypt Depth	Muscular Thickness	V/C ^†^
Con	0.89 ± 0.04	0.23 ± 0.10	0.40 ± 0.02	3.90 ± 0.31
PSMS5	0.92 ± 0.04	0.24 ± 0.01	0.40 ± 0.03	3.86 ± 0.31
PSMS10	0.95 ± 0.04	0.23 ± 0.01	0.37 ± 0.02	4.23 ± 0.17
PSMS15	1.08 ± 0.09	0.21 ± 0.01	0.35 ± 0.02	4.17 ± 0.75
PSMS20	1.16 ± 0.13	0.20 ± 0.02	0.32 ± 0.03	5.73 ± 0.72
*p*-value	0.149	0.259	0.154	0.080

^1^ P.SMS replaced WPCS at levels of 0%, CON; 5%, PSMS5; 10%, PSMS10; 15%, PSMS15; or 20%, PSMS20; * Height, depth, and thickness in mm; ^†^ V/C, Villus height/Crypt depth.

**Table 5 animals-14-03416-t005:** Serum biochemical indices of sheep in five groups before and after the feeding of P.SMS.

Times	Groups ^1^	Serum Biochemical Indices *				
		ALT ^†^ U/L	AST U/L	ALP U/L	TP g/L	ALB g/L
Before (1d)	Con	18.82 ± 1.74	102.70 ± 15.17	214.67 ± 30.30	71.75 ± 2.91	28.13 ± 2.13
	PSMS5	21.10 ± 1.83	108.72 ± 1.28	244.76 ± 25.63	70.82 ± 1.94	32.38 ± 0.26
	PSMS10	22.82 ± 2.69	92.88 ± 1.91	186.14 ± 20.96	76.30 ± 1.00	30.74 ± 1.83
	PSMS15	16.23 ± 2.32	109.50 ± 2.52	275.83 ± 9.86	69.80 ± 1.15	33.73 ± 0.30
	PSMS20	19.53 ± 1.55	107.95 ± 5.24	241.60 ± 22.94	74.70 ± 2.19	32.83 ± 0.68
After (90d)	Con	18.10 ± 1.27 ^b^	88.80 ± 18.12	161.90 ± 11.32	70.83 ± 3.87	27.87 ± 2.88
	PSMS5	27.00 ± 6.50 ^a^	96.70 ± 5.10	266.25 ± 42.65	70.90 ± 1.70	33.30 ± 0.50
	PSMS10	21.80 ± 1.22 ^ab^	95.30 ± 4.29	191.25 ± 29.06	70.76 ± 1.70	30.61 ± 0.66
	PSMS15	14.46 ± 3.53 ^ab^	76.10 ± 17.75	156.74 ± 43.66	72.03 ± 3.53	25.58 ± 4.50
	PSMS20	21.70 ± 1.42 ^b^	107.67 ± 3.97	265.05 ± 40.69	67.08 ± 1.64	31.28 ± 0.98
Effects ^‡^	Times	0.549	0.100	0.305	0.148	0.194
	Treatments	0.012	0.536	0.183	0.686	0.249
	Times &Treatments	0.571	0.447	0.274	0.255	0.294

^1^ P.SMS replaced WPCS at levels of 0%, CON; 5%, PSMS5; 10%, PSMS10; 15%, PSMS15; or 20%, PSMS20; * ALT, Alanine aminotransferase; AST, Aspartate Aminotransferase; ALP, Alkaline phosphatase; TP, Total Protein; ALB, Albumin; ^†^ The labels ‘a’ and ‘b’ in the same row indicate significant differences between different groups (*p* ˂ 0.05). The absence of labels indicates no significant difference; ^‡^ Impact of different factors on indicators: numerical values less than 0.05 typically indicate statistical significance, suggesting a strong likelihood that the dependent variable will change due to the factor’s influence.

## Data Availability

The raw datasets generated during the current study are available in the NCBI (PRJNA1085887) repository.

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
