# Peer review of "Effects of Spent Substrate of Oyster Mushroom (Pleurotus ostreatus) on Feed Utilization and Liver Serum Indices of Hu Sheep from the Perspective of Duodenal Microorganisms"

_animals, 2024, doi:10.3390/ani14233416_

Round 1

Reviewer 1 Report

Comments and Suggestions for Authors

The present study mainly determined the effects of Pleurotus ostreatus spent mushroom substrate (P.SMS) on feed utilization and liver serum indices using the perspective of duodenal microorganisms in 3-month-old Hu lamb. The result showed that dietary P.SMS supplementation influenced the apparent digestibility of feed, in which 10% P.SMS induced change in structure of the duodenal community, and had no side-effect on liver healthy. The result of this study shows that P.SMS is safe when the additive amount less than 15%, providing theoretical basis for the appropriate use of spent mushroom substrate in ruminant. 

However, there are several flaws in the manuscript. 

Overall, the research only focused on the duodenal microorganisms and serum liver serum in Hu lamb supplemented with P.SMS, the growth performance of sheep in five group should be evaluated to further explain the effect of P.SMS replaced with some WPCS in sheep ration. Moreover, gut microbial diversity analysis at genus level needs to pay attention to reveal the composition difference among different groups.    

In details:

Line 66-69: References are needed. Please describe the difference between the earlier study mentioned here and the present study.

Line 114: The middle sentence is obscure.

Line 119-120: The temperature and humidity should be described.

Table 2: Whole Corn silage should be replaced with WPCS according to the context.

Line 110-120: The feeding scale should be shown in this part.

Line 135: The replicate is not mentioned in the experimental design.    

Line 199-201: Please explain the relationship between the previous study and the present study, or adding the reference. It is hard to know the experimental animal, feed, treatment, and so on, in the previous study.

Line 201: Supplementary Figure 1 B is not cited in the text in the present paper.

Line 229-231: The sentence is obscure.

Line 240, 243-244: OTUs in the sentence should be replaced with “unique OTUs”.

Line 247-250: Bacterial species name at the phylum level should be presented in italic type or not.

Line 268: A full stop should be added at the end of this sentence.

Line 278-282: Bacterial species name at the phylum level should be presented in italic type or not.

Figure 6: Based on the bacterial species name, LEfSe analysis of bacterial and fungal samples were seems to conducted at different level including family, geuns, species, etc.

Line 335: What the mean of DMI is.

Line 367: The sentence is obscure.

Line 415-416, 430-422: There sentences should be placed in the Discuss part, rather than in Conclusions.

Others:

The data presentation form in Tables and Figures is average ± SD or SEM.

The format in the second and third title should be in consistency in the whole paper.

The figures in the paper is fuzzy and blurry.

Comments on the Quality of English Language

On the whole, the English language is this paper is easy to understand. Some sentence need to be revised and polished.

Reviewer 2 Report

Comments and Suggestions for Authors

The author and his colleagues, explored the effect of waste flat mushroom substrate on feed utilisation, serum indicators and duodenal intestinal flora of lake sheep, the study has some innovative and practical application value, waste utilisation and degradation of farming costs. However, the writing, drawing normality and language representation of the article need to be further optimised so that readers can just understand the article, the specific problems are as follows:

1. What is the reference standard for the setting of the additive ratio?

2. Figure1 is presented in a table to make the comparison clearer to the reader.

3. Is the diet a complete mix? What is the dry matter content?

4. Representative pictures of duodenal addition slices

5. The phylum level is too large, add a compositional abundance plot at the family or genus level.

6. Rows 261-263, no significant difference in a diversity, why is it dropping again?

7. All the authors show is at the gate level, too unspecific from a gut microbial perspective, needs to be added to the discussion.

8. Line 159, remove Stool.

9. What is the threshold for adding LDA to the material approach?

Comments on the Quality of English Language

The English could be improved to more clearly express the research.

Reviewer 3 Report

Comments and Suggestions for Authors

Dear Editor and Authors,

The authors examined the effects of replacing whole-plant corn silage in the diet with Pleurotus ostreatus spent mushroom substrate on feed utilization and liver serum indices in Hu sheep, as well as its impact on duodenal microorganisms. The study topic is quite important in terms of evaluating a waste material in the diet. The study is quite detailed, and the authors' approach to the topic is very satisfying. I believe there needs to be a revision in the last paragraph of the introduction. Some methods have been omitted in the Materials and Methods section. I think a revision is necessary in the Results section regarding serum biochemical indices. While the Discussion section is satisfactory, some explanations contradict the tables presented in the Results section. These contradictions need to be addressed. Additional information and my evaluation report are provided below.

Best regards

Comments:

In the text, the abbreviation for Pleurotus ostreatus spent mushroom substrate is presented in two different forms: P.SMS and PSMS. Consistency should be maintained.

L47: I think the word "generations" is not suitable in this context. It is rarely preferred with this meaning in scientific articles, so please express it differently.

L47-68, L87: The examples and explanations provided in lines L47-68 and L87 are not related to the physical properties of PSMS; rather, they pertain to the different evaluation methods of PSMS. Therefore, it is recommended to change the term "physical properties." The expression "physical properties" is more commonly used for cases involving particle size or when the feed is provided in forms such as powder, pellet, or crumble.

L84-86: Please provide a reference.

L87-94: I believe this section is written in a rather confusing way. Instead of first explaining what was done in this study and then stating the purpose, I think it would improve the quality of your work to start by presenting the hypothesis of your study, followed by describing what was done to test this hypothesis, and finally emphasizing which gap in the literature your findings will address.

L111: Please specify whether you used SD or SEM.

L110-112: Since the term "replication" is mentioned in the Sample Collection and Processing section, please specify the number of replications for each group in this part.

L114: Please use “consisted” instead of “PSMS10onsisted”.

L114-115: Please provide a reference (NRC or another source).

L182: Please add an explanatory paragraph about tissue morphology in the Materials and Methods section.

L212-213: To indicate a trend, a p-value between 0.05 and 0.10 is generally preferred. In your study, a trend can only be mentioned between villus height and crypt depth. Therefore, it is recommended to revise this statement accordingly.

L218: Please check the abbreviation for V/C.

L222: You mentioned that you collected blood samples at the beginning and end of the study to determine serum biochemical parameters, and you have visualized the results separately for the beginning and end of the study. However, it is not possible to demonstrate the additive effect of PSMS in the diet in this way over a 90-day feeding period. Therefore, the figure and the corresponding statements need to be revised to include interaction effects as well.

L333-336: It would be better to provide the p-values in Supplementary Figure 1 so that we can determine whether there is a trend.

L342-344: In Table 2, it is understood that as the PSMS ratio in the diets increases, the ME values of the diets also increase. Although this indicates that the ME value of PSMS is higher than that of whole-plant corn silage, how did you make such a conclusion? This statement contradicts the values in Table 2.

Round 2

Reviewer 1 Report

Comments and Suggestions for Authors

Line 90-94: Reference [13] showed negative effect of P.SMS on production, rumen indices, and several serum indices in Hu sheep. The positive effects of P.SMS mentioned in the former sentence should be presented to explain the necessity of the present research.

Line 231: The sentence is obscure.

Line 244: Figure 1. is not cited in the text.

Line 272-273: Is it correct or not to cite Figure 3A and 3B here?

Line 285: More details should be described about Neocallimastigomycota here.

Line 288-290: More details should be described about Candidatus_Saccharimonas in bacteria and Aspergillus and Trichoderma in Fungi here.

Line 323: The sentence is obscure.

Line 333-341: There are five groups in Figure5(A), while only two groups in Figure5(D), why?

Line 418: The time information should be added here.

Line 420-423: Consider the difference in dose of SMS between these two paper.

Line 431-432: The sentence is obscure.

Line 440: Reference(s) related with previous authors should be cited.

Line 445-448: Which phylum is mentioned here?

Line 424-458: In order to consist with the result 3.5.1 and 3.6, the bacterial and fungal with significant abundance should be mainly discussed in this paragraph.

Others:

The format in the second and third title, as well as Figures should be in consistency in the whole paper.

The samples number for Serum Biochemical Indexes, Duodenum histomorphology, and 16S rRNA gene sequencing should be added in “2. Materials and Methods”.

Reviewer 3 Report

Comments and Suggestions for Authors

Dear Editor and Authors,

The revised article has shown significant improvement compared to the original version. The shortcomings I noted in my previous reviewer report have been addressed by the authors. Therefore, I believe this revised article is scientifically publishable. I congratulate the authors for this valuable work and hope to have the opportunity to review and read many more of their excellent studies in the future.

Best regards.

Author Response

Dear reviewer,

Thank you for your approval of our experiments and revisions, as well as for your detailed explanation of the revision process. Your feedback has brought to our attention important issues that we had not previously considered. We will address these concerns in our future work and strive to conduct even better experiments.

Best regards.